# Management of Anesthesia and Perioperative Procedures in a Child with Glucose-6-Phosphate Dehydrogenase Deficiency

**DOI:** 10.3390/jcm11216476

**Published:** 2022-10-31

**Authors:** Ana Cicvarić, Josipa Glavaš Tahtler, Tea Vukoja Vukušić, Ivančica Bekavac, Slavica Kvolik

**Affiliations:** 1Department of Anesthesiology, Resuscitation, and ICU, Osijek University Hospital, 31000 Osijek, Croatia; 2Faculty of Medicine, University of Osijek, 31000 Osijek, Croatia; 3Department of Pediatric Surgery, Osijek University Hospital, 31000 Osijek, Croatia

**Keywords:** glucose-6-phosphate dehydrogenase (G6PD), oxidative stress, perioperative management, anesthesia management, postoperative monitoring

## Abstract

Glucose-6-phosphate dehydrogenase (G6PD) is an enzyme that helps red blood cells work properly; it participates in the production of antioxidants and helps to defend cells against oxidative damage. With all this in mind, patients with G6PD deficiency may be very sensitive and vulnerable to different oxidative stressors, because they can cause some serious medical conditions of which hemolytic anemia is common in adults and severe jaundice in newborns. The most common triggers of hemolysis in G6PD deficiency are infections, medications, metabolic conditions such as diabetic ketoacidosis, hypothermia, and a very important item—surgical stress. During the operative period, the anesthetic goal is to reduce stress and monitor if the hemolysis occurs, and of course, treat it if it occurs. In our case, the combination of sevoflurane inhalation anesthesia with the addition of sufentanil proved to be safe and effective in the management of a child with G6PD deficiency.

## 1. Introduction

The most common human enzyme deficiency in the world is glucose-6-phosphate dehydrogenase (G6PD) deficiency [1,2]. It is an inherited X-linked disorder, so it most often affects men. G6PD is an enzyme that helps red blood cells work properly, it participates in the production of antioxidants and helps to defend cells against oxidative damage. G6PD is an oxireductase. It is included in the first step in the pentose phosphate pathway in which nicotinamide adenine dinucleotide phosphate (NADPH) is created. NADPH provides reduced glutathione inside the cell and is important because glutathione acts as an antioxidant and protects cells from oxidative damage.

Most of the cells have some extra system of other metabolic pathways that can generate the necessary intracellular NADPH, but red blood cells do not have that. With all this in mind, patients with G6PD deficiency may be very sensitive and vulnerable to different oxidative stressors, because they can cause some serious medical conditions of which hemolytic anemia is common in adults and severe jaundice in newborns. Breakdown of the red blood cells can be caused by infections, medications, stress, or different food such as fava beans [3,4].

Based on the functional severity of G6PD enzyme deficiency, we distinguish five classes of diseases.

Class I G6PD deficiency is present and is associated with chronic non-spherocytic hemolytic anemia. Class II variants have less than 10% enzymatic activity, but no chronic nonspherocytic anemia is present. Class III variants have a residual enzymatic activity of 10 to 60%. Class IV variants have normal enzyme activity, and class V variants have increased G6PD activity. In some cases, symptoms of anemia can develop very fast, and they may require emergency medical assistance [3,4].

Drugs capable to induce hemolysis are non-steroidal anti-inflammatory drugs (NSAIDs), anticonvulsants, sulfonamides, nitrofurantoin and chloramphenicol, diuretics, insulin, ranitidine and thiopentone. An in vitro study showed that anesthetic medications unsafe to use in classes I, II, and III of G6PD are isoflurane, sevoflurane, and diazepam [5].

Here, we will present the management of perioperative procedures and anesthesia in a child of class III of G6PD enzyme deficiency.

## 2. Case Report

The patient was a sixteen-month-old boy who was admitted to the hospital for elective left testicular retention surgery.

He was born by the vaginal route at 36 + 3/7 weeks of gestation, birth weight was 2200 g, birth length was 47 cm, and Apgar scores 10/10. It was his mother’s second, regularly monitored pregnancy in which oligohydramnios was diagnosed. A few hours after delivery, he developed hyperbilirubinemia, after which his general condition worsened. Diagnostics were started in the tertiary institution and the lack of G6PD enzyme was confirmed. It was recommended to avoid fava beans and all other beans in the diet and medications that are contraindicated due to deficiency of the G6PD enzyme in red blood cells. Moreover, after birth, the child received a blood transfusion several times due to the development of anemia. He takes an oral iron preparation and vitamin D3. All the time the patient is under the multidisciplinary supervision of a neuro pediatrician, physiatrist, and pediatric surgeon. He was monitored by a pediatric surgeon because of bilateral testicular hydrocele and left testicular retention that needed to be operated and that was his first surgery. Perioperative he was assessed as ASA physical status III.

He had good perioperative laboratory findings (Table 1). The last blood transfusion was three months before surgery. He had no allergy reported by the day of the surgery. His weight was ten kilograms. After careful evaluation, it was decided that the anesthesia technique for this procedure and this patient would be general anesthesia. Before the beginning of surgery, peripheral venous access had been established. Standard monitoring was set which included pulse oximetry, electrocardiography, noninvasive arterial blood pressure monitoring, and expiratory carbon dioxide (ETCO2). A heated blanket was placed on the operating table and the operating room was heated too.

The patient was preoxygenated with 100% oxygen through a face mask, after which was applied atropine 0.2 mg and sufentanyl 3 mcg. Anesthesia was administrated by the inhalation technique, along with oxygen 40%, nitric oxide 60%, and sevoflurane 6%–3%–2%. After the patient was asleep and when the depth of anesthesia was sufficient, I-gel 1.5 was placed. He received another 2 mcg of sufentanyl during the procedure. The procedure itself lasts 45 min, and it was orchidopexy of the left side according to Shoemaker, hernioplasty of the left side according to Ferguson, adhesiolysis of the frenulum of prepuce, and penis frenectomy. The child was hemodynamically and respiratory stable during the whole procedure. After the end of surgery and after awakening, he was briefly monitored at the post-anesthesia care unit (PACU). At the PACU he was stable, he needed no further analgesics. He was discharged to the Department of Pediatrics surgery. His postoperative laboratory finding was also good (Table 1), and he had no need for any transfusions. At the Department of Pediatric surgery after approximately five hours postoperatively, he was a little bit grumpy and he pointed to the site of the operation site. According to the Faces scale, his pain was rated as a pain level of 4 and he was given ibuprofen suppository, after which he calmed down and fell asleep. After 24 h, he was discharged to home care to his parents.

## 3. Discussion

Here, we have presented anesthesiology procedures in a sixteen-month-old boy with known glucose-6-phosphate dehydrogenase deficiency. Anemia, jaundice, hemoglobinuria, fatigue, and back pain are the most common symptoms of acute hemolysis associated with G6PD deficiency [6]. These are consequences of G6PD deficiency, that may lead to hemoglobin denaturation and subsequent hemolysis. The most common triggers of hemolysis in G6PD deficiency are infections, medications, metabolic conditions such as diabetic ketoacidosis, hypothermia, and a very important item—surgical stress [5,6,7,8]. With the perioperative procedures presented here, these triggers were controlled, and complications of hemolysis were avoided as the patient had good preoperative laboratory findings and his general condition was good without signs of infection. Treatment of infection before elective surgery is very important because, during active infections, inflammatory neutrophils, as well as the use of certain antibiotics can induce hemolysis in patients with G6PD deficiency [7]. Therefore, with these children, it is necessary to take information about the child’s general condition from the child’s parents. A detailed clinical examination of the child, which includes an examination of the skin, oral cavity, and auscultation, with minimal laboratory indicators of the absence of inflammation, should also be conducted before surgery.

During anesthesia and in the postoperative period, drugs involved in causing oxidative stress and hemolysis should be avoided, and surgical stress should be reduced using appropriate anxiolytics and analgesics [7].

Concomitant use of diazepam or midazolam with sevoflurane or isoflurane should be avoided as in vitro study has shown that they have an inhibitory effect on G6PD and this drug combination may increase the severity of hemolysis. Ketamine does not cause G6PD inhibition and can be used as an intravenous anesthetic for G6PD deficiency [9]. However, some other reviews consider sevoflurane and isoflurane safe to use in G6PD-deficient patients [7,10,11]. Our patient did not receive diazepam preoperatively as he was not disturbed and restless. For induction, sevoflurane was used, but it did not lead to hemolysis, and it did not endanger the stability of the patient. An important factor for reducing surgical stress is adequate pain management, which in our patient was achieved intraoperatively with sufentanyl, with monitoring of respiratory and hemodynamic parameters that were stable throughout the procedure. In children, the use of paracetamol in adequate doses is recommended for postoperative pain, but it is not considered a safe drug in patients with G6PD deficiency, and it was not used in our patient either. Given that the patient postoperatively had a pain intensity of 4 according to the FACE scale, he was given an ibuprofen suppository, after which the pain was adequately controlled according to his clinical signs—he calmed down and fell asleep [12]. Regional anesthesia can also be performed in children, but it is usually combined with general anesthesia or used as a method of postoperative pain control [13,14]. Considering the specificity of patients with G6PD deficiency, attention should be given to the choice of local anesthetic, because for instance lidocaine and prilocaine are not recommended, while bupivacaine is shown to be safe to use [15,16].

Postoperative monitoring includes markers of hemolysis and, if it occurs, timely treatment is necessary. In this child, there were no indicators of hemolysis, such as jaundice, anemia, heart rhythm changes on the EKG, blood pressure changes, or dyspnea. Therefore, in the above case, a normal blood count was sufficient laboratory confirmation of the child’s postoperative recovery (Table 1). In a situation where any of these clinical signs would appear in the postoperative period, it would be necessary to perform more extensive laboratory diagnostics to confirm hemolysis. Therefore, for anesthetic procedures, it is important to take all preventive measures to avoid possible hemolysis. Our patient was monitored for the next 24 h, during which he showed no deterioration in his condition, and he was discharged home.

## 4. Conclusions

During the operative period, the anesthetic goal is to reduce oxidative stress and monitor if the hemolysis occurs, and of course, treat it if it occurs. The use of anesthetics such as midazolam, sevoflurane, and isoflurane in patients with G6PD deficiency is controversial. In our case, the combination of sevoflurane inhalation anesthesia with the addition of sufentanil proved to be safe and effective in the management of a child with G6PD deficiency.

## Figures and Tables

**Table 1 jcm-11-06476-t001:** Laboratory findings.

	Perioperative Laboratory	Postoperative Laboratory
WBC [×10^9^/L]	6.9	7.2
RBC [×10^12^/L]	3.13	2.93
Hgb [g/L]	103	101
Hct [L/L]	0.314	0.317
MCV [fL]	100.3	108.2
MCH [pg]	32.9	34.5
MCHC [g/L]	328	319
Plt [×10^9^/L]	323	406

(WBC—white blood count, RBC—red blood cells, Hgb—hemoglobin, Hct—hematocrit, MCV—mean corpuscular volume, MCH—mean corpuscular hemoglobin, MCHC—mean corpuscular hemoglobin concentration, Plt—platelet count).

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
