# Peer review of "Management of Anesthesia and Perioperative Procedures in a Child with Glucose-6-Phosphate Dehydrogenase Deficiency"

_jcm, 2022, doi:10.3390/jcm11216476_

Round 1

Reviewer 1 Report

In this manuscript, the authors presented that the management of anesthesia and perioperative procedures, that is, the combination of sevoflurane inhalation with the addition of sufentanil, proved safe and effective in managing the child with G6PD deficiency. The aim and scope of the study are well explained. The abstract is easy to understand. The study is quite comprehensive, however; some concerns must be attended

1     The Introduction. Information about G6PD is too brief and may be difficult for readers who are not familiar with G6PD enzyme action and deficiency in the metabolic pathway to understand the rest of the paper.

2     Introduction, Line 32-36. The information is repetitive with the abstract. It must rewrite.

3     The abbreviations of the table need to be spelled in order to understand the paper.

Author Response

Dear Sir/Madam

after you send us the Review Report, we read it and made the necessary changes. 

In the introduction, the line 32-36 is rewriten. Also, we correct the abbreviations of the table. It is also added section on regional anesthesia in children and the safety of local anesthetics in patients with G6PD deficiency.

Best regards

Reviewer 2 Report

Dear authors,

I was pleased to read your paper entitled "management of anesthesia and perioperative procedures in a child with G6PD).

The case report appears well-described, unambiguous, and concise.

The pathology you described is common and appears helpful in identifying a safe anesthetic path.

I would only supplement with a brief examination of why you did not choose neuraxial anesthesia if associated with sedation, and on the management of post-operative analgesia.

Probably the use of subarachnoid or other analgesic treatments could be considered oversized for the type of post-operative pain that the young patient would face, but I think it is helpful to express an opinion also on the appearance of local anesthetics.

As an edition note, the type of paper above the title is missing.

Author Response

(The authors gave the same response as above.)
